# Is There a Dose-Response Relationship between Acute Physical Activity and Sleep Length? A Longitudinal Study with Children and Adolescents Living in Sweden

**DOI:** 10.3390/children8090808

**Published:** 2021-09-15

**Authors:** Alexandra Nordman, Marita Friberg, Yvonne Forsell

**Affiliations:** 1Department of Global Public Health, Karolinska Institutet, 17177 Solna, Sweden; alexandra.nordman@stud.ki.se; 2Department of Living Conditions and Lifestyles, Public Health Agency of Sweden, 17165 Solna, Sweden; Marita.Friberg@folkhalsomyndigheten.se

**Keywords:** exercise, physical activity, sleep, children, adolescents, Sweden

## Abstract

A declining physical activity (PA) and sleep in children and adolescents have been observed during the previous decades. PA could benefit sleep, but the findings are mixed. The aim of the present study was to examine if there is a dose-response relationship between time spent in acute moderate and vigorous physical activity (MVPA) and sleep length in children and adolescents. Additional aims were to examine if the sleep length is higher for children and adolescents who conduct at least an average of 60 min in MVPA/day and to study differences between sex and school years. The study population consists of 262 participants in school year 5 (aged 11 years), 7 (aged 13 years), and 9 (aged 15 years). Accelerometers measured MVPA while sleep diaries measured sleep length. A linear and longitudinal mixed effect linear regression was conducted to study the primary aim. The secondary aims were studied with linear regressions. Included confounders were sex, school year, school stress, screen time, menstruation onset, family household economy, and health status. A stratified regression for sex and school year was conducted. The linear regression showed no statistically significant findings in the crude or adjusted model. The stratified linear regression found a significant positive association for girls but a negative association for school year 5. No associations were found in the longitudinal regression or when comparing sleep length for participants that did and did not spend an average of at least 60 min in MVPA/day. A dose-response relationship was found in the stratified linear regression, implying a possible weak association. The statistically non-significant differences between participants that did and did not spend an average of at least 60 min in MVPA/day implies that spending an average of at least 60 min in MVPA/day may not be associated with a higher mean sleep length.

## 1. Introduction

The years in childhood and adolescence are a foundation for the rest of your life. Health and well-being are important for development and factors, like good sleep habits and adequate levels of physical activity (PA), are important [1,2]. PA is associated with physical [3], mental [3], social and cognitive health benefits [4], and physical inactivity increases the risk for ill-health [3]. The benefits include lower adiposity, improved cardio-metabolic biomarker profile, increased physical fitness, improved bone health, and improved motor skill development [4]. The PA recommendations from the World Health Organization (WHO) for children and adolescents are to spend an average of 60 min/day in moderate-to-vigorous PA (MVPA), conduct musculoskeletal strengthening exercises at least three days/week, and limit sedentary time [3,5]. PA can be divided into chronic PA (regular PA) and acute PA (sporadic occurrences). As opposed to chronic PA, which has long-term effects, the effects of acute PA on sleep are detectable only on the night following the acute PA [6]. For instance, a meta-analysis including adolescents and adults found that the effects of acute PA resulted in 10 min of significantly increased sleep length, a deeper sleep, and falling asleep faster on nights following a day with acute PA [6].

Around the globe, physical fitness muscle power and cardiovascular fitness decreased in children and adolescents between the year 1986 to 2015 [7]. Nowadays, a majority (81%) of the children and adolescents spend less than 60 min in MVPA every day [8].

It is common that PA declines during adolescence to early adulthood [2]. Girls spend less time in PA than boys and since the year 2001, the prevalence of insufficient PA has decreased for boys but not for girls around the globe [8]. In Sweden, a majority of children and adolescents do not meet the PA recommendations [8]. In Sweden, the prevalence of children and adolescents that meet the PA recommendations have not increased during the previous decades [3] and a majority spends less than 60 min every day in PA (82.2% boys and 87.3% girls) [8].

PA can be measured both with subjective measurements like questionnaires and other self-reported measures and via objective measures like doubly labeled water and accelerometers [2]. The most common method for measuring PA is via objective measures with accelerometers, and accelerometers provide more robust and precise measures of the PA than subjective measures [2,9].

Sleep is important for children and adolescents’ physical, psychological, and academic functioning [2], and insufficient sleep have been associated with ill-health [4,10,11]. In addition, insufficient sleep could increase psychological ill health and threaten academic success [4,11]. A previous systematic review found that children and adolescents with longer sleep durations had lower adiposity indicators, better emotional regulation, higher academic achievements, and better life quality than children with shorter sleep duration [4]. While there are no sleep recommendations in Sweden, The National Sleep Foundation in the United States has developed national sleep recommendations for children and adolescents; between 9–11 h of nocturnal sleep for children aged 6–13 years and 8–10 h for 14–17-year-olds [12]. It is common that the sleep patterns shift during adolescence and this shift might be caused by psychological, social, behavioral, and pubertal changes [2]. The physical changes during adolescence and puberty include changes in the circadian rhythm and the homeostatic sleep components, which influence the sleep-wake cycle, sleep timing, sleep duration, and sleep architecture [2].

During the past decades, there has been a trend of declining sleep length in children and adolescents around the globe, and sleep deprivation has increased [4,13]. The decline can be seen in most countries (also in Sweden), age groups, and both sexes [13,14]. In addition, it is more common that boys have later nocturnal sleep times and do not meet the recommendations than girls [14].

Previous studies that have examined the association between PA and sleep have resulted in mixed findings and potential mechanisms are still mainly unknown [11,15]. A previous systematic review with children and adolescents resulted in contradictory findings regarding sleep length and time in MVPA [15]. However, the authors concluded that sleep length might be associated with sedentary behavior rather than time in MVPA [15]. Furthermore, previous studies have examined if and how PA could affect sleep length, but the studies have been unable to detect statistically significant findings [2,10,11]. A common belief is that conducting PA shortly before sleep could disrupt sleep. A prior study found mixed results regarding this belief; PA had potentially negative effects on sleep in adults but positive effects on children [2]. Despite this, the existing literature implies that PA probably is beneficial for sleep in children, adolescents, and young adults [2,10,16]. For instance, a systematic review found that children and adolescents with more PA were more likely to experience better nocturnal sleep [2]. In addition, an intervention study found that increased time in MVPA improved sleep duration, sleep depth, sleep efficiency, increased rapid eye movement (REM) sleep latency, and decreased sleep onset latency in children and adolescents [2,10,17]. However, due to the age-related decline in both PA and sleep during adolescence [1,2] less than 20% of the children and adolescents meet both the PA and sleep recommendations [1]. In addition, those with low socioeconomic positions and poor economies are less active and have a higher risk of short sleep length [5,18]. Screen time has also been associated with a shorter sleep length [14] and increased time in physical inactivity [19]. The theoretical concepts consist of hypotheses for how PA could affect sleep and includes physical [2,10,20] and psychological [20] changes. Physical changes include the facts that sleep serves energy conservation, body restoration, and thermoregulatory functions—which are factors that are reversely affected by PA [2,10]. The psychological changes from PA are decreased anxiety and stress, which could improve sleep [20]. Opdal et al. aimed to examine the association between PA and mental distress with a longitudinal study design [21]. However, Opdal et al. were unable to detect changes in the PA and mental distress during the follow-up period [21]. Though additional studies are needed, it is possible that the relationship between PA and sleep is synergetic and works via multiple complex physiological and psychological pathways [2]. 

In addition to the potential confounders of sex, age, mental health, socioeconomic position, and screen time that have been mentioned above, the association between PA and sleep might also be mediated or moderated by sleep quality, daylight length, and the PA characteristics [2,10].

The evidence of an association between PA and sleep in children and adolescents are limited and this depends partly on methodological differences [2], poor designs [2,11], and limited populations in the reported studies [4]. There is a need for additional studies which have the potential to examine an association and yield high-quality evidence [1]. It is the first time that measurements on sleep have been included and analyzed using the Health Behavior in School aged Children (HBSC) survey questionnaire by the Public Health Agency of Sweden. To the knowledge of the authors, an association between PA and sleep has not previously been studied in children and adolescents living in Sweden. 

### Aim and Research Questions

The primary aim was to examine if there is a dose-response relationship between time spent in acute MVPA and sleep length in children and adolescents. The secondary aim was to examine if the sleep length was higher for participants that spent an average of at least 60 min in MVPA/day. An additional aim was to distinguish differences in PA and sleep length between sex and school years.

## 2. Materials and Methods

### 2.1. Study Design and Population

The study population included children and adolescents in the middle (5th) and upper (7th and 9th) stage of compulsory school in Sweden and are thus aged 11, 13, and 15 years. A flowchart of the number of participating schools and pupils can be seen in Figure A1 in Appendix A. As an international WHO study conducted every four years, the Public Health Agency of Sweden conducts the HBSC questionnaire survey to monitor the health and lifestyle behaviors in school-aged children. The most recent HBSC survey was conducted in 2017/18, and during that survey, an additional sub-study was conducted. For the HBSC sub-study, Statistics Sweden selected a random sample of 80 schools, which included 3600 pupils in the school years 5, 7, and 9. Schools with <100 pupils and schools where only one school class could be recruited were excluded. In a second step, a random selection of 2 classes per school was conducted. From the 80 schools that were invited, 20 accepted, 41 declined, and 19 were not available (response rate 33%). The survey was conducted during one whole week between March and June 2018, and 16 of the 20 schools that accepted participation completed the data collection. All pupils in the randomly selected school classes were invited to participate, resulting in 39 classes and 825 pupils; 11 classes in school year 5, 17 classes in year 7, and 11 classes in year 9. An invitation letter was sent to the pupils’ guardians.

### 2.2. Data Collection

The HBSC sub-study included a smaller random sample of schools and school classes, as described in the section above, than the original HBSC survey but included more measures than the survey questionnaire. The additional measures consisted of PA and sleep length measurements during one whole week. The Swedish School of Sport and Health Sciences (GIH) was responsible for collecting the sleep and PA measurements, which they recorded with sleep diaries and accelerometers. The accelerometer “ActiGraph GT3X” (ActiGraph LCC, Pensacola, FL, USA) was used. Within the time period of March to June 2018 and during the one whole week of the school’s data collection, the participating pupils were asked to wear the accelerometer on the hip during daytime and on the left wrist during nighttime. Furthermore, the participants were asked to record the time they went to bed at night and awoke in the morning in the sleep diaries. Of the 825 pupils in the study population, 378 pupils both wore the accelerometer, recorded their sleep, and answered the questionnaire. The HBSC survey was conducted as a government assignment (S2017/01227/FS). Ethical assessment has been made by the Public Health Agency of Sweden (04145-2017).

### 2.3. Exposure and Outcome

The Romanzini counts/minute was used to define the PA intensities using the following cut-offs; sedentary time (<720 counts/minute), low-intensity PA (720–3025 counts/minutes), and MVPA (>3025 counts/minute). Accelerometer non-wear time was defined as 0 counts/minute for > 60 min and was excluded from the analysis to reduce misclassification and ensure a valid representation.

Sleep length was determined from the sleep diaries. In the sleep diaries, the participants recorded the date, hour, and minute (5 min intervals) when the participant awoke in the mornings and turned off the lights in the evenings. The available choices to record the hour the participant awoke was 05:00–12:00 in the morning. The choices to record the hour when the participant turned off the light was 19:00–03:00 in the evening. The participants’ recordings of the time they turned off the light in the evenings and the time when they awoke were translated into the sleep lengths (hours).

Met sleep recommendation was defined using the recommendations from the United States National Sleep Foundation; for school year 5 at least 9 h/night and for school year 7 and 9 at least 8 h/night.

#### Confounders

Information about confounders was collected from the HBSC questionnaire, which was answered by the child. The included confounders were sex, school year, school stress, screen time, menstruation onset, family household economy, and health status. Regarding school stress, the question was; “How stressed do you feel about your school work and the answers were categorized into; no perceived school stress (“not at all” and “somewhat”), perceived school stress (“pretty much” and “much”). Regarding screen time, the variable consisted of three questions; “How many hours/day do you spend in front of the TV, video (including Youtube or similar), DVD and other entertainment on a screen”, “How many hours/day do you usually spend on a computer, game console, tablet computer, smartphone or other electronic devices (do not count games with movements like Wii Fitness, Dance central or Sports Champions on Wii, Xbox, KINECT or Playstation Moves)” and “How many hours/day do you usually spend on electronic devices like a computer, computer tablets or smartphones for other purposes than for example school homework, email, tweeting, Facebook, chat and surf on the internet?”. The answers were categorized into; “0–1 h/day on 1–3 devices/social media”, “2–4 h/day on 1 device/social media”, “2–4 h/day on 2 devices/social medias”, “2–4 h/day on 3 devices/social medias”, “5–7 h/day on 1 device/social media”, “5–7 h/day on 2 devices/social medias” and “5–7 h/day on 3 devices/social medias”. The question regarding the family’s household economy was “How well off economically do you think your family is?”, The following categories were used; good (“very good” and “good”), moderate (“mediocre”) and bad (“not so good” and “not good at all”). The question regarding the health status was “How do you think your health is?”, and the following categories were used; good (“very good“ and “good“) and bad (“pretty bad“ and “bad”).

### 2.4. Statistical Analysis

Before conducting the statistical tests for a potential dose-response relationship, the Akaike Information Criterion (AIC) test and Wald test was conducted. The AIC test uses maximum-likelihood estimation to evaluate how the chosen model fits with the data, the smaller the AIC value, the better the fit [22]. The AIC test was used to study if a potential dose-response relationship between sleep length and time in MVPA was best described as linear or quadratic. Furthermore, the AIC test was also used to test if the variable “time in MVPA” was best defined as continuous or categorical (categories for the quintiles of time in MVPA). The Wald test is a significance test that also can test linear hypotheses [23]. A Wald test was conducted after the AIC test to test if the relationship between sleep length and time in MVPA departs from linearity. 

To detect a dose-response relationship between acute MVPA and sleep length, both a linear regression and a longitudinal mixed effect linear regression was used. A linear regression was used to detect an association between the mean time spent in MVPA and the mean sleep length [24]. A longitudinal mixed effect linear regression was conducted to analyze if the MVPA during the day was associated with the sleep length the following night [25]. A descriptive table containing the longitudinal values of sleep length and PA among, between, and within participants was created before the longitudinal regression. A mixed model was applied to account for both fixed- and random effects that are present when studying the association between sleep and PA [26]. A linear regression was conducted to analyze the potential association between sleep length and spending an average of at least 60 min in MVPA/day. For all regressions, a crude, adjusted, and stratified (for sex and school year) analysis was conducted.

## 3. Results

### 3.1. Demographic Information

The demographic information is described in Table 1, and an extended demographic table can be found in Appendix A (Table A1). The analysis includes 262 participants with sleep and MVPA measurements for four days and over 500 min accelerometer wear-time/day. The majority of the participants were girls, born in Sweden, in school year 7, reported good health and family household economy, and perceived little or no school stress (70.2%). Furthermore, most participants reported a screen time category of two to four h/day on three different devices or social media (27.0%), and 58.8% of the girls had gone through the onset of menstruation.

#### Physical Activity and Sleep Length

The sleep length distribution can be seen in Figure A2 and in Appendix A and the variation in sleep length and time spent in MVPA between the measured days can be seen in Figure A3 and Figure A4 in Appendix A. For every additional measurement day, the sleep length increased significantly with 0.09 h (*p* = 0.007, CI: 0.012–0.165) (Figure A5 in Appendix A). The mean sleep length and time in MVPA are presented in Table 2. A more detailed table of how the sleep length, accelerometer wear-time, time in MVPA, and frequency and percentage of met sleep recommendation varied with each measurement day can be found in Appendix A (Table A2). Participants in school year 5 had a mean sleep length of 9.22 h/night varying between 9.07–9.43 h. Participants in school year 7 had a mean sleep length of 8.68 h/night varying between 8.46–8.94 h. Participants in school year 9 have a mean sleep length of 8.28 h/night varying between 7.90–8.32 h. Furthermore, the prevalence of meeting the sleep recommendation varied between the measured days; between 60.8–70.1% for school year 5, between 66.7–74.8% for school year 7, and between 48.6–62.9% for school year 9 (Table A2 in Appendix A). Without accounting for the school year, the mean sleep length was similar for boys (8.88 h/night) and girls (8.87 h/night). However, the prevalence of the boys that met the sleep recommendation varied between 59.0–72.3% during the measurement days, and for the girls between 69.0–75.5% (Table A2 in Appendix A). 

The distribution of the mean time spent in MVPA/day can be seen in Figure A6 in Appendix A. Time spent in MVPA decreased statistically non-significantly with 1.42 min/measurement day (*p* = 0.185, CI:−3.522–0.681) (Figure A7 in Appendix A). Participants in school year 5 had a MVPA mean time of 103.68 min/day varying between 98.46–118.82 min, school year 7 mean time of 76.70 min/day varying between 73.06–91.77 min, and school year 9 65.37 min/day (Table 2). However, only about half of the participants in school year 9 had recordings of their sex (n = 19), and the mean time in MVPA was higher when not defining the sex (n = 35); 75.32 min/day varying between 73.06–91.77 min (Table A2 in Appendix A). Based on four days of measurements, the prevalence of the participants that spent an average of at least 60 min in MVPA/day was 78.1% and decreased with school year; 87.9% in school year 5, 75.6% in school year 7, and 57.1% in school year 9. Boys spent approximately 10 more mean minutes in MVPA in school years 5 and 7 respectively than girls. However, in school year 9, girls spend approximately 3 more mean minutes in MVPA than boys. The prevalence of participants that spent an average of at least 60 min in MVPA/day was somewhat lower for boys (75.9%) than girls (77.4%).

### 3.2. Characteristics of the Relationship

The AIC test resulted in a smaller value for the linear relationship (AIC = 742.04) than for the quadratic relationship (AIC = 743.61). The data fits best when time spent in MVPA was defined as a continuous variable rather than a categorical variable (AIC = 743.13). However, the Wald-test implied that the relationship between sleep length and time in MVPA was not fully compatible with a simple linear regression (*p* = 0.561). This needs to be remembered in the following analyses.

### 3.3. Dose-Response of Mean Time in MVPA and Sleep Length

The crude and adjusted linear regression of the mean sleep length and time in MVPA found no association (Table 3). However, in the stratified linear regression (sex and school year), the mean sleep length varied with mean time spent in MVPA for girls and school year 5. When stratifying for sex, every additional minute spent in MVPA increased the sleep length in girls with 0.01 h (*p* = 0.035, CI = 0.000–0.011) (Translated to 0.6 additional mean minutes of sleep for every additional mean minute spent in MVPA), no association was seen in boys (Figure 1). When stratifying for school the year, the mean sleep length varied with time in MVPA, every additional minute spent in MVPA resulted in a 0.01 h decrease in the mean sleep length for participants in school year 5 (*p* = 0.041, CI = −0.010–0.000) (Figure 2) (Translated to a decrease of 0.6 mean minutes of sleep for every additional mean minute spent in MVPA). No statistically significant associations were found for school years 7 and 9. The stratified analysis for school year showed that the mean sleep length decreased significantly with school year; year 7 had 0.56 h less mean sleep (*p* = 0.000, CI = −0.820–−0.307), and year 9 had 1.05 h less mean sleep (*p* = 0.000, CI = −1.412–−0.690), compared to school year 5. Participants that perceived school stress had a lower mean sleep length compared to participants that did not (0.47 h less mean sleep, *p* = 0.001, CI = −0.738–−0.196). Girls that had gone through the menstruation onset had a lower mean sleep length compared to girls that had not (0.66 h less mean sleep, *p* = 0.000, CI = −0.994–0.330). However, when adjusting for the school year, the association disappeared. In addition, participants in the highest screen time category had a lower mean sleep length compared to those in the lowest category (0.99 h less mean sleep, *p* = 0.013, CI = −1.768–−0.216). 

### 3.4. The Longitudinal Dose-Response Relationship

The sleep length and time spent in MVPA among, between, and within participants is described in Table 4. The mean sleep length among the participants was 8.82 h/night. The mean time spent in MVPA was 90.44 min/day.

The longitudinal mixed effect linear regression showed that the sleep length did not vary with time in MVPA, either in the crude (0.00 h, *p* = 0.892, CI = −0.002–0.002) (Figure 3) or adjusted model (0.00 h, *p* = 0.083, CI = −0.055–0.000). In the stratified analysis for sex, the analysis showed that sleep length did not vary with time in MVPA either for boys (0.00 h, *p* = 0.832, CI = −0.004–0.003), or girls (0.00 h, *p* = 0.784, CI = −0.003–0.002). Similarly, the stratified analysis for the school year showed that the sleep length did not vary with time in MVPA, either for participants in school year 5 (0.00 h, *p* = 0.120, CI = −0.004–0.001) or school year 7 and 9 (0.00, *p* = 0.462, CI = −0.004–0.002).

### 3.5. The Association between Met and Unmet Sleep Recommendation and Time in MVPA

No differences in the mean sleep length were detected when comparing participants that did and did not spend an average of at least 60 min in MVPA/day. According to the crude model, the mean sleep length was 8.762 h/night for participants that did not spend an average of at least 60 min in MVPA/day. Though not statistically significant, the mean sleep length was 0.065 h longer for participants that spent at least 60 min in MVPA/day (*p* = 0.656, CI = −0.222–0.352). The analysis remained statistically non-significant in the adjusted model (*p* = 0.376).

## 4. Discussion

### 4.1. The Major Findings in Regards to the Aims

No statistically significant dose-response relationship between sleep length and time in MVPA was found when applying a linear or a longitudinal mixed effect linear regression. However, a statistically significant dose-response relationship was found in the stratified linear regressions for sex and school year. Girls had a positive linear dose-response relationship, and participants in school year 5 had a negative dose-response relationship. These results imply that there may be a weak dose-response relationship between time in MVPA and sleep length. Though, the quality of these results is deemed as poor because the dose-response relationship was solely found in the stratified linear regression for the mean time in MVPA and sleep length and not in the longitudinal model that regards temporality. The results are in line with the literature where a systematic review found that multiple cross-sectional studies have found an association between PA and sleep in adolescents [2] but only a few longitudinal studies have found a statistically significant association [17,20,27]. No differences were detected in the sleep lengths and time spent in MVPA among, between, or within participants. The mean sleep length did not differ when comparing participants that did and did not spend an average of at least 60 min in MVPA/day. These results imply that spending an average of at least 60 min in MVPA/day, measured objectively, may not be associated with a higher mean sleep length.

### 4.2. Implication of Findings and Alternative Explanations

The sleep length did not vary with time in MVPA either in the crude or adjusted linear regression, only in the stratified linear regressions for sex and school year. The reason for this might be that the association between time in MVPA and sleep length may differ between the participants’ characteristics. Previous studies have acknowledged that the effect that PA possibly has on sleep may be moderated by characteristics of the individual and the PA [2,10]. These studies have discussed that age, sex, fitness level, sleep quality, BMI, PA intensity, PA duration, PA environment, and the time of day that the PA is conducted may moderate the association [2,10]. In the stratified linear regression, a positive association was found for girls while a negative association was found for school year 5, and no association was found for boys and school year 7 and 9. Both the crude and adjusted model analyzed the participants as one group. However, since the association may differ with the participants’ characteristics and have different directions, the non-stratified models might not be able to detect an association.

The stratified linear regression for sex and school year showed that the mean sleep length varied with the mean time in MVPA, however, the sleep length did not vary with time in MVPA in the longitudinal mixed effect linear regressions. A reason for this could be that four days of measurements were not enough, or that the results have been tainted by the residual confounder of chronic PA. A previous study concluded that four days of measurements correlate well with seven days of measurements [9]. However, it is possible that four days of measurements was not enough to detect significant findings. Another explanation for the non-significant findings from the crude and adjusted linear regression and longitudinal mixed effect linear regression could be the unmeasured variable of chronic PA. Acute and chronic PA can result in similar sleep benefits, but it may be important to distinguish between them when studying the association with sleep [6]. The chronic PA and its effect on sleep occur when an individual has high PA over a long time period (PA intensities between 35–80% of max), and the effects on sleep could last during a longer time period [2,4,10]. Chronic PA is strongly associated with a high fitness level, longer sleep duration, and improved sleep quality [6]. Furthermore, chronic PA has a larger effect on sleep than acute PA, and the effect of the acute PA could be masked on individuals with chronic PA [6]. Unfortunately, regular PA was not measured in the present study and chronic PA could therefore not be adjusted. Chronic PA could have caused underestimations of the results because the effect from the acute PA may be masked by effects from the chronic PA. 

Other explanations for the non-significant findings in the longitudinal mixed effect and linear regressions could be that PA affects other sleep outcomes than sleep length or that other intensities than MVPA affects sleep length. Some studies have included multiple aspects of sleep and many results suggest that PA has a positive effect on several sleep outcomes [2,6,10,17,20,27,28]. The optimal PA dose might vary depending on the health outcome, and it is unclear which dose of PA affects sleep [4]. Furthermore, the WHO PA recommendation of an average of 60 min MVPA/day is based on and applicable to subjectively measured PA. It is not yet established how much time of objectively measured MVPA is applicable to the health benefits and prevention of ill-health that have been associated with the WHO PA recommendations.

Another consideration regarding the non-significant findings in the longitudinal mixed effect and linear regressions is that the non-significant findings might be an effect of small changes in the PA and sleep lengths. In similarity to the discussion in the article by Opdal et al., the lack of significant findings might be caused by small changes in the exposure and outcome from the baseline to follow-up [21]. A greater variation in the time spent in PA and sleep lengths might have resulted in significant findings.

Finally, the current study explored a linear dose-response relationship between time spent in MVPA and sleep length. A linear dose-response relationship was applied because the AIC-test implied a better fit with the data with a linear relationship rather than a quadratic relationship. However, there may exist non-linear associations between time in MVPA and sleep length that have not been explored in the current study. For instance, two studies that examined a potential association between screen time and sleep length in children and adolescents resulted in different conclusions regarding the association’s linearity. Both studies examined a linear and quadratic relationship, but while Hisler et al. concluded that the association was, most often, best described by a quadratic relationship [29] Twenge et al. found that the association was fairly well suited to a linear relationship [30]. Thus, when studying the association between time in MVPA, and other health exposures, and sleep length, it may be important to first explore which model fits best with the data. 

### 4.3. Sex and School Year Differences

A majority of the participants were girls (65.1%), but no large sex difference in time in MVPA or sleep length was detected. One explanation for the uneven sex distribution might be that girls were more consistent with wearing the accelerometer for the instructed amount of wear-time and measurement days compared to boys. Boys spent approximately 10 more mean minutes in MVPA in school years 5 and 7 than girls, while the opposite was seen in year 9 (girls 3 more mean minutes than boys). However, the prevalence of the participants that spent an average of at least 60 min in MVPA/day based on four days of measurements was somewhat lower for boys (75.9%) than girls (77.4%). In comparison to the present study, previous studies have concluded that the prevalence that spends at least 60 min in MVPA every day is higher for boys than girls [3,8]. It is possible that the sex differences in MVPA are not large when studying an average of objectively measured PA within a measurement period, but the difference may be large when comparing each day within a measurement period.

Similarly, a previous study from Sweden reported that 84.7% of the adolescents do not conduct at least 60 min of PA every day [8], and the results in the present study imply that 78.1% of the children and adolescents conducts an average of at least 60 min of MVPA/day. The discrepancies are probably caused by the self-reported MVPA in the mentioned study and by the objective and average measured MVPA in the present study. 

The mean sleep length was similar for boys and girls, however, the prevalence that met the sleep recommendation was somewhat lower for boys (59.0–72.3%) than for girls (69.0–75.5%) and boys were a little less likely to have met sleep recommendations. The prevalence that met the sleep recommendation varied between 60.8–70.1% for school year 5, between 66.7–74.8% for year 7, and between 48.6–62.9% for year 9, and this may be a cause of concern for the health and well-being. Furthermore, on all the measured days, the mean sleep length was above the recommended level for all school years except for school year 9. Both the prevalence of the participants that met the sleep recommendation and that spent an average of at least 60 min in MVPA/day decreased with the school year, and the prevalence was lowest for the participants in school year 9. The age-related decline is in line with previous findings where PA has been shown to decrease in the transition to higher school years [1,3]. Another interesting observation was that the prevalence of the participants that spent an average of at least 60 min in MVPA/day was higher than the prevalence that met the sleep recommendation. However, based on previous studies with self-reported data, it seems like children and adolescents living in Sweden are more probable to have a sufficient sleep length [8,14] than to spend sufficient time in physical activity [8]. One reason for this discrepancy could be that participants that participate in studies with objectively measured PA may be more physically active than participants that participate in studies with subjectively measured PA. Furthermore, it is possible that the participants in the above-mentioned study have underestimated the time they spend in PA or that the participants in the current study have underestimated their sleep lengths.

### 4.4. Strengths

The major strengths of this study include the use of accelerometers and the nationwide random selection of schools [9]. After excluding participants with less than four days of measured PA and with 500 min of wear-time/day, the remaining wear-time was high (between 726.56–908.19 min/day). By applying a nationwide random selection, the generalizability of the findings to the target population increased, and biases were limited.

### 4.5. Limitations

Selection bias is probably present both when selecting schools and when schools and pupils accepted participation. Schools with less than 100 pupils and schools where only one school class could be recruited were excluded from the random selection of schools. It is possible that the selection did not include schools in rural areas and schools in urban areas with different orientations. In similarity, the pupils that participated may be more physically active and health-conscious than pupils that did not participate. The selection bias could have caused an overestimation of the results. It is likely that the subjectively measured sleep length has caused non-differential outcome misclassification since it is less accurate and could overestimate the sleep length [4]. Additionally, other aspects of sleep were not included such as falling asleep faster or having less interrupted sleep. Findings from a systematic review indicated that it takes less time to fall asleep and that the awake time after the sleep onset decreased after an acute PA exposure [2]. Differential outcome misclassification could have caused both an under or overestimation of the results depending on if the sleep length was under or overestimated by the participants. Accelerometers have some weaknesses in the ability to reliably register some type of activities, for instance, biking. In addition, the accelerometer must be taken off when conducting water sport activities. Thus, it is possible that the time spent in MVPA and the prevalence of participants that spent an average of at least 60 min in MVPA/day might have been misclassified and caused underestimations. The limitations of accelerometer measures may be accounted for in future studies by adding subjective measures of time in MVPA to the accelerometer measures and thereafter study the correlation between these measures. Some confounders have been adjusted for in the analysis, however, the results may be biased by residual confounders such as chronic PA, body mass index, socioeconomic position, the timing of PA, and consumption of caffeinated beverages. 

### 4.6. Generalizability

It is possible that the results have been affected by selection bias and thus caused low external validity. It has previously been concluded that the accelerometer finding from the sub-survey are generalizable to the study population of the HBSC questionnaire survey (825 pupils) [3]. However, participants with accelerometer measures were excluded if they did not have four days of accelerometer measures and sleep diary recordings and 500 min wear time/day. Thus, even if the accelerometer measures are generalizable, it cannot be assumed that the findings are generalizable either to the HBSC questionnaire survey or in a wider context. 

### 4.7. Public Health Relevance

The results showed that every additional mean minute spent in MVPA either can increase or decrease the mean sleep length by 0.01 h depending on different characteristics. It is possible that the effect varies with other characteristics than sex and age. Time spent in MVPA and a sleep length within the recommendations yield multiple health benefits and if PA and sleep are associated, there could be an interaction that might result in further health benefits. Even if an effect of 0.01 h is not large, it could be valuable to have the knowledge of a potential effect on sleep when implementing interventions and policies. Interventions aimed at increasing the PA in groups where the PA already is high might have negative effects on the sleep lengths, and on the contrary, interventions aimed at increasing the PA in groups where the PA is low might have positive effects on the sleep lengths. Another public health relevance is that there was a large decline in the time spent in MVPA- and the prevalence that spent an average of at least 60 min in MVPA/day with school years. More interventions are needed to increase the PA in adolescents.

### 4.8. Future Research

To increase the quality of the evidence for an association, future studies could apply a longitudinal study design with the ability to assess temporality. The quality of the evidence could also be improved by intervention studies with random assignment of PA exposures. If a longitudinal mixed effect linear regression is applied to analyze a dose-response relationship in future studies, a suggestion is to use more than four days of measures. Additionally, objective measures of sleep, as well as multiple sleep outcomes, should be applied, for instance, heart rate monitors. In addition, the chronic PA should probably be adjusted for. Future research should also examine the upper and lower bounds of PA that potentially affect sleep.

## 5. Conclusions

There may be a weak dose-response relationship between time spent in MVPA and sleep length and the direction of the relationship may vary with characteristics of the individual and the PA. Further studies are needed to confirm the association. Furthermore, spending an average of at least 60 min in MVPA/day may not be associated with a higher mean sleep length. Lastly, no large differences in sleep lengths or time in MVPA between boys and girls were found, however, MVPA declines with the school year. 

## Figures and Tables

**Figure 1 children-08-00808-f001:**
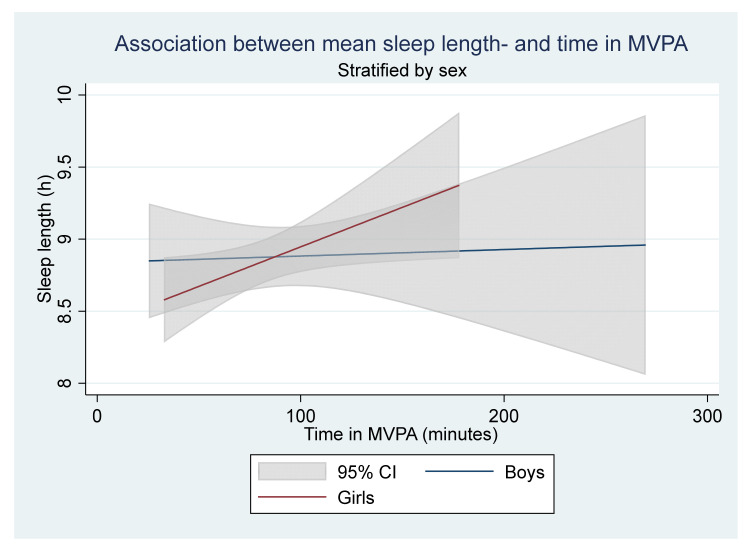
The stratified linear regression for sex. A statistically significant linear association was found for girls (red line, the steeper incline) but not boys (blue line, smaller incline). The analysis includes 235 participants (152 girls and 83 boys).

**Figure 2 children-08-00808-f002:**
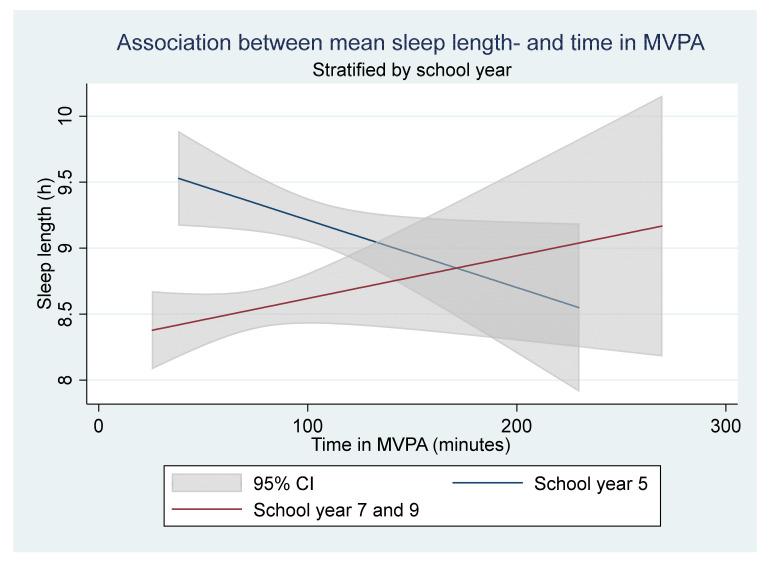
The stratified linear regression for school year. A statistically significant linear association was found for school year 5 (blue declining line) but not for school year 7 and 9 (red inclining line). The analysis includes 262 participants (106 in school year 5 and 156 participants in school year 7 and 9).

**Figure 3 children-08-00808-f003:**
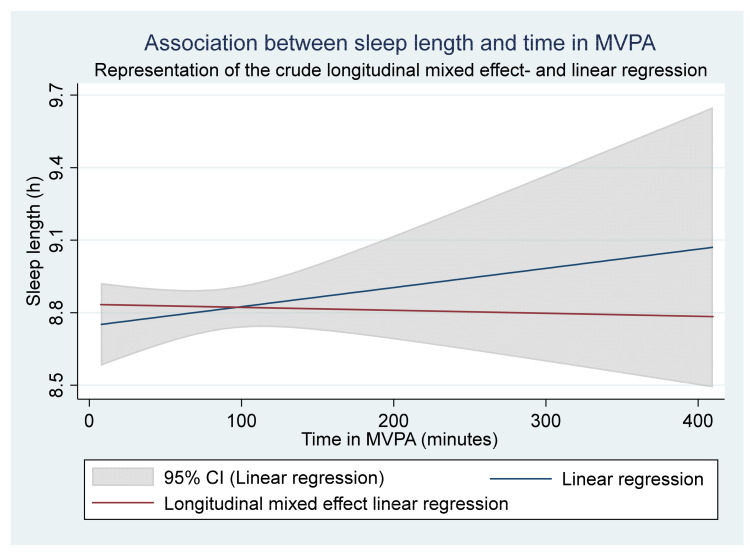
Graphic presentation of the longitudinal mixed effect regression (red line, small declining line (0.00 h/minute in MVPA)) and the linear regression (blue line, increasing line (0.00 h/minute in MVPA). The analysis includes 262 participants.

**Table 1 children-08-00808-t001:** Population demographic information of sex, school year, family household economy, health status and participant’s county of birth of the 262 participants.

Variable	Frequency (n)	Percent (%)
Sex		
Boys	83	34.9
Girls	152	65.1
Total ^#^	235	100.0
School year		
5	106	30.4
7	121	46.4
9	35	13.2
Total	262	100.0
Family household economy		
Good	191	79.9
Moderate	43	18.0
Bad	5	2.1
Total^#^	239	100.0
Health status		
Good	222	92.9
Bad	17	7.1
Total ^#^	239	100.0
Participant’s country of birth		
Sweden	218	91.2
Norway, Denmark, Finland or Island	1	0.4
Europe	7	2.9
Outside Europe	12	5.0
Do not know	1	0.4
Total ^#^	239	100.0

^#^ The number of participants is lower due to incomplete response to questions.

**Table 2 children-08-00808-t002:** The average sleep length and time spent in moderate- and vigorous physical activity (MVPA) across school year and sex for the 235 participants with records on sex (83 boys, 152 girls, 100 participants in school year 5, 116 in school year 7 and 19 in school year 9).

Variable	School year			Total
	5	7	9	
*Time spent in MVPA (minutes)*				
Boys				
Mean	111.94	81.30	63.96	94.44
SD	37.37	39.67	33.72	41.48
n	39	38	6	83
Girls				
Mean	98.49	74.52	66.03	83.40
SD	32.05	21.97	33.25	29.98
n	61	78	13	152
Total				
Mean	103.68	76.70	65.37	87.25
SD	34.65	28.87	32.46	34.74
n ^#^	100	116	19	235
*Sleep length (hours)*				
Boys				
Mean	9.17	8.71	8.17	8.88
SD	0.89	0.98	0.61	0.95
n	39	38	6	83
Girls				
Mean	9.25	8.67	8.33	8.87
SD	0.86	0.99	0.76	0.97
n	61	78	13	152
Total				
Mean	9.22	8.68	8.28	8.88
SD	0.87	0.98	0.70	0.96
n ^#^	100	116	19	235

* Standard Deviation; ^#^ The number of participants is lower due to incomplete response to question about sex.

**Table 3 children-08-00808-t003:** The crude and adjusted model for the linear regression (n = 262).

Association between Mean Sleep Length and Time in MVPA *	Change in Sleep Length (Hours) for Each Minute in MVPA *	SE^#^	*p*-Value	Confidence Interval (95%)
Crude model	0.00	0.00	0.074	0.000–0.007
Adjusted model	0.00	0.00	0.471	−0.008–0.004

* Moderate- and vigorous physical activity; ^#^ Standard error.

**Table 4 children-08-00808-t004:** The mean sleep length and time spent in moderate and vigorous physical activity (MVPA) among, between, and within the participants. The analysis included 262 participants with 4 days of measurements (1048 observations).

Variable	Mean ^#^	Standard Deviation	Minimum	Maximum	Numbers of Observations (n)
Sleep length (hours)					
Among participants	8.82	1.42	1.50	14.75	1048
Between participants		0.97	5.48	11.71	262
Within participants		1.04	1.32	13.44	4
Time in MVPA (minutes)					
Among participants	90.44	47.59	7.50	409.75	1048
Between participants		33.52	21.06	284.75	262
Within participants		33.83	−69.32	243.81	4

^#^ The mean sleep length and time in MVPA varies from Table 2 (8.88 h and 87.25 min) due to incomplete response to questions about sex.

## Data Availability

The data presented in this study are not publicly available due to ethical reasons.

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
