# Peer review of "Is There a Dose-Response Relationship between Acute Physical Activity and Sleep Length? A Longitudinal Study with Children and Adolescents Living in Sweden"

_children, 2021, doi:10.3390/children8090808_

Round 1

Reviewer 1 Report

The study idea is smart.

Abstract; please report participants’ (not: pupil) age and gender-ratio; not all readers are familiar with the Swedish schooling system.

General comment:

Given that – overall – results were spurious, I strongly recommend to thoroughly inspect the data set for non-linear correlations. In doing so, the authors might find associations, which at a linear level are non-existent. To get an impression, I strongly recommend to giving a very close look at (Hisler et al., 2020; Twenge et al., 2020; Twenge and Campbell, 2019; Twenge and Martin, 2020; Twenge et al., 2018). Twenge et al could identify non-linear associations between a broad variety of relevant health dimensions among adolescents. Further studies with relevant non-linear associations are: (Alizadeh et al., 2021; Kalak et al., 2015; Mahlmann et al., 2017; Shimizu et al., 2019).

Zero-results between psychological functioning, sleep and physical activity longitudinally among adolescents: (Opdal et al., 2019).

A further issue is related to the assessment of physical activity patterns with non-GPS-assisted devices and the rules/recommendations to wear an actigraph. To illustrate: A 10-years old girl going to school by bike for 50’ has a lower physical activity-record, compared to her comrade walking to school for 10’/day, simply because the actigraph does nor register any movement. Likewise, a 10-years old girl swimming at a semi-professional level 5x/week for 50-90’ per swimming session will have a lower physical activity level, again when compared to her comrade walking to school for 10’/day, simply because the swimmer is not allowed to wear any items, including watches, bracelets, necklaces, and… actigraphs. As such, from a purely scientific point of view, I do not think that actigraphs are reliable devices. Next, to fill the gap of missing values, others (Hatzinger, M. et al., 2010; Hatzinger, Martin et al., 2010; Hatzinger et al., 2014) reported that actigraph values were double-checked with diary-assisted records and information. Or simply put: Actigraph-derived data are always as reliable as the quality of the diaries completed in parallel. To further illustrate this issue, it appeared that the authors of the present study had the issue that 378 out of 825 participants (45.8%) provided reliable data, or the other way around: More than 50% of their participants were unable or unwilling to provide reliable and “objective” data.

As mentioned above, please note that (Opdal et al., 2019) were unable to find any associations between physical activity patterns and psychological well-being.

Given this background, I suggest the following procedures:

#1; thoroughly integrate the results of Opdal et al., 2019;

#2; restart your plot

#3; thoroughly inspect the scatter plots for non-linear associations.

#4; consider to report subjective physical activity reports, compared to “objective” actigraphically based data.

I’m aware that my comments might be frustrating, though, I hope that the authors take the suggestions with a sporty spirit.

References

Alizadeh, M., Delborde, Y., Ahmadpanah, M., Seifrabiee, M.A., Jahangard, L., Bazzazi, N., Brand, S., 2021. Non-linear associations between retinal nerve fibre layer (RNFL) and positive and negative symptoms among men with acute and chronic schizophrenia spectrum disorder. Journal of psychiatric research 141, 81-91.

Hatzinger, M., Brand, S., Perren, S., Stadelmann, S., von Wyl, A., von Klitzing, K., Holsboer-Trachsler, E., 2010. Sleep actigraphy pattern and behavioral/emotional difficulties in kindergarten children: association with hypothalamic-pituitary-adrenocortical (HPA) activity. Journal of psychiatric research 44(4), 253-261.

Hatzinger, M., Brand, S., Perren, S., Stadelmann, S., von Wyl, A., von Klitzing, K., Holsboer-Trachsler, E., 2010. Sleep actigraphy pattern and behavioral/emotional difficulties in kindergarten children: association with hypothalamic-pituitary-adrenocortical (HPA) activity. Journal of psychiatric research 44(4), 253-261.

Hatzinger, M., Brand, S., Perren, S., Von Wyl, A., Stadelmann, S., von Klitzing, K., Holsboer-Trachsler, E., 2014. In pre-school children, sleep objectively assessed via actigraphy remains stable over 12 months and is related to psychological functioning, but not to cortisol secretion. Journal of psychiatric research 55, 22-28.

Hisler, G., Twenge, J.M., Krizan, Z., 2020. Associations between screen time and short sleep duration among adolescents varies by media type: evidence from a cohort study. Sleep Med 66, 92-102.

Kalak, N., Brand, S., Beck, J., Holsboer-Trachsler, E., Wollmer, M.A., 2015. Association between subjective actual sleep duration, subjective sleep need, age, body mass index, and gender in a large sample of young adults. Neuropsychiatric disease and treatment 11, 107-113.

Mahlmann, L., Gerber, M., Furlano, R.I., Legeret, C., Kalak, N., Holsboer-Trachsler, E., Brand, S., 2017. Impaired objective and subjective sleep in children and adolescents with inflammatory bowel disease compared to healthy controls. Sleep Med 39, 25-31.

Opdal, I.M., Morseth, B., Handegård, B.H., Lillevoll, K., Ask, H., Nielsen, C.S., Horsch, A., Furberg, A.-S., Rosenbaum, S., Rognmo, K., 2019. Change in physical activity is not associated with change in mental distress among adolescents: the Tromsø study: Fit Futures. BMC public health 19(1), 916-916.

Shimizu, M., Gillis, B.T., Buckhalt, J.A., El-Sheikh, M., 2019. Linear and Nonlinear Associations between Sleep and Adjustment in Adolescence. Behavioral sleep medicine, 1-15.

Twenge, J.M., Blake, A.B., Haidt, J., Campbell, W.K., 2020. Commentary: Screens, Teens, and Psychological Well-Being: Evidence From Three Time-Use-Diary Studies. Front Psychol 11, 181.

Twenge, J.M., Campbell, W.K., 2019. Media Use Is Linked to Lower Psychological Well-Being: Evidence from Three Datasets. Psychiatr Q 90(2), 311-331.

Twenge, J.M., Martin, G.N., 2020. Gender differences in associations between digital media use and psychological well-being: Evidence from three large datasets. Journal of adolescence 79, 91-102.

Twenge, J.M., Martin, G.N., Campbell, W.K., 2018. Decreases in psychological well-being among American adolescents after 2012 and links to screen time during the rise of smartphone technology. Emotion (Washington, D.C.) 18(6), 765-780.

Reviewer 2 Report

In the current paper, the authors investigate whether there is a relationship between physical activity and sleep length in children. Although it is mainly a negative finding, this information is clearly interesting and may be a piece of a puzzle to better understanding. 

In general, the paper is clearly written. I have a few minor points.

  • line 17 and in methods e.g. line 125: please mention the age of the children. A reader who is not familiar with the school system in Sweden, has no idea about age.
  • lines 135-more: the authors talk about 'one consecutive' week. It is not clear to me what is meant, since in my few consecutive always is someting in a row, that means more than one.
  • lines 135-more: a month Mars does not exist, probably this should be March
  • lines 166-167: in my view it should be 'turned off' the lights
  • line 279; this is the answer to the main question, isn't it? So, in my view this result should not be put in an appendix
  • line 319 table 3: this table is not clear to me. The legend states 'difference in ....' but I only see one mean sleep lenght and one mean MVPA plus a range. Please also explain the abbreviations in legends of tables and figures and improve wording.
  • line 335-336; if this is a result worthwhile to mention, then please give data and statistics.
  • lines 339-342: the sentences are not complete, please rephrase and indicate what was tested, what was found etc. Only mentioning the statistical test is not informative.
  • line 345; typo - linear
  • lines 432-435: how to explain the different results? Could the authors please elaborate on this point a bit?

Author Response

Thank you for the suggested improvements!

All the points made by the reviewer have now been changed in the article, in accordance with the reviewers comments. 

Round 2

Reviewer 1 Report

The authors did a great job in revising the manuscript.

“Changing the plot” means; you have to rebuild the whole story in the Introduction section, because you were unable to find a robust association between MVPA and sleep lengths. Relatedly, the Introduction section needs a more thorough description of those studies in line with your data, and those studies not in line with your data. Here Opdal et al will help you, along with Twenge et al (Twenge et al., 2020).

A further issue is that it appears that you acknowledge that accelerometer measurements do not appear to be reliable, though, this fact appeared to have no impact on the overall handling of the method and data.

“Subjectively measures physical activity often yields in over- and underestimated physical activity and as a sole measure, subjectively measured physical activity is not very trustworthy.” This statement is critical in two ways; First, the authors do not provide evidence for this statement; accordingly, it appears a personal and thus subjective point of view. Second, the claim is in sharp contrast of what has been published before in the field of research (Craig et al., 2003; McKeon et al., 2019; Rosenbaum et al., 2020; Rosenbaum and Ward, 2016; Schilling et al., 2018); as such, published data do not support your claim; given this, once again, in my opinion, the authors have to rebuild both the Introduction and Discussion section.

“…. longitudinal study design [20]. Unfortunately, Opdal et al were unable to…”; I suggest to delete “unfortunately”; this for two reasons: First, specifiers such as “unfortunately” are discouraged; second, in this specific case, the authors of the present manuscript should be rather “fortunate”, as the pattern of results of Opdal et al. sustains your data.

References

Craig, C.L., Marshall, A.L., Sjöström, M., Bauman, A., E.,, Booth, M.L., Ainsworth, B.E., Pratt, M., Ekelund, U., Yngve, A., Sallis, J.F., Oja, P., , 2003. International physical activity questionnaire: 12-country reliability and validity. Medicine and science in sports and exercise 35, 1381-1395.

McKeon, G., Steel, Z., Wells, R., Newby, J.M., Hadzi-Pavlovic, D., Vancampfort, D., Rosenbaum, S., 2019. Mental health informed physical activity for first responders and their support partner: a protocol for a stepped-wedge evaluation of an online, codesigned intervention. BMJ open 9(9), e030668.

Rosenbaum, S., Morell, R., Abdel-Baki, A., Ahmadpanah, M., Anilkumar, T.V., Baie, L., Bauman, A., Bender, S., Boyan Han, J., Brand, S., Bratland-Sanda, S., Bueno-Antequera, J., Camaz Deslandes, A., Carneiro, L., Carraro, A., Castañeda, C.P., Castro Monteiro, F., Chapman, J., Chau, J.Y., Chen, L.J., Chvatalova, B., Chwastiak, L., Corretti, G., Dillon, M., Douglas, C., Egger, S.T., Gaughran, F., Gerber, M., Gobbi, E., Gould, K., Hatzinger, M., Holsboer-Trachsler, E., Hoodbhoy, Z., Imboden, C., Indu, P.S., Iqbal, R., Jesus-Moraleida, F.R., Kondo, S., Ku, P.W., Lederman, O., Lee, E.H.M., Malchow, B., Matthews, E., Mazur, P., Meneghelli, A., Mian, A., Morseth, B., Munguia-Izquierdo, D., Nyboe, L., O'Donoghue, B., Perram, A., Richards, J., Romain, A.J., Romaniuk, M., Sadeghi Bahmani, D., Sarno, M., Schuch, F., Schweinfurth, N., Stubbs, B., Uwakwe, R., Van Damme, T., Van Der Stouwe, E., Vancampfort, D., Vetter, S., Waterreus, A., Ward, P.B., 2020. Assessing physical activity in people with mental illness: 23-country reliability and validity of the simple physical activity questionnaire (SIMPAQ). BMC Psychiatry 20(1), 108.

Rosenbaum, S., Ward, P.B., 2016. The Simple Physical Activity Questionnaire. Lancet Psychiatry 3(1), e1.

Schilling, R., Schärli, E., Fischer, X., Donath, L., Faude, O., Brand, S., Pühse, U., Zahner, L., Rosenbaum, S., Ward, P.B., Carraro, A., Gerber, M., 2018. The utility of two interview-based physical activity questionnaires in healthy young adults: Comparison with accelerometer data. PLoS One 13(9), e0203525.

Twenge, J.M., Blake, A.B., Haidt, J., Campbell, W.K., 2020. Commentary: Screens, Teens, and Psychological Well-Being: Evidence From Three Time-Use-Diary Studies. Front Psychol 11, 181.

Author Response

Please see the attached file for the authors response to the reviewers comments.
